# Magnitude of underweight, wasting and stunting among HIV positive children in East Africa: A systematic review and meta-analysis

Biruk Beletew Abate[1]*, Teshome Gebremeskel Aragie[2], Getachew Tesfaw[1]

1 Department of Nursing, College of Health Sciences, Woldia University, Woldia, Ethiopia, 2 Department of Anatomy, College of Health Sciences, Woldia University, Woldia, Ethiopia

☯ These authors contributed equally to this work.
* birukkelemb@gmail.com

**Data Availability Statement:** All relevant data are within the paper and its Supporting information files.

## Abstract

### Background

Malnutrition on the background of HIV (Human Immunodeficiency Virus) infection is a complex medical condition that carries significant morbidity and mortality for affected children, with greater mortality from SAM (Severe Acute Malnutrition) among HIV-positive children than their HIV-negative peers. HIV-induced immune impairment heightened risk of opportunistic infection and can worsen nutritional status of children. HIV infection often leads to nutritional deficiencies through decreased food intake, mal-absorption and increased utilization and excretion of nutrients, which in turn can hasten death.

### Objective

The aim of this systematic review and meta-analysis was to assess the magnitude of underweight, wasting and stunting among HIV positive children in East Africa.

### Methods

The authors systematically reviewed and meta-analyzed studies that assessed the prevalence of underweight, wasting and stunting among HIV positive children in East Africa from PubMed, Cochrane Library, Google Scholar, and Gray Literatures using PRISMA (Preferred Reporting Items for Systematic Reviews and Meta-analyses) guideline. The last search date was December 30/2019. The data was extracted in excel sheet considering country, study design, year of publication, prevalence reported. Then the authors transformed the data to STATA 14 for analysis. Heterogeneity across the studies was assessed by the Q and the $I^2$ test. A weighted inverse variance random-effects model was used to estimate the magnitude of underweight, wasting and stunting. The subgroup analysis was done by country, year of publication, and study design. To examine publication bias, a funnel plot and Egger's regression test were used.

**Funding:** The authors received no specific funding for this work

**Competing interests:** The authors have declared that no competing interests exist.

**Abbreviations:** DHS, Demographic Heath Survey; WHO, World Health Organization; CI, Confidence interval; AOR, Adjusted odds ratio; SAM, Severe Acute Malnutrition; PRISMA, Preferred Reporting Items for Systematic Reviews and Meta-Analyses.

## Results

For the analysis a total of 22 studies with 22074 patients were used. The pooled prevalence of under-weight, wasting, and stunting among HIV positive children in East Africa was found to be 41.63% (95%CI; 35.69–47.57; $I^2$ = 98.7%; p<0.001), 24.65% (95%CI; 18.34–30.95; $I^2$ = 99.2%; p<0.001), and 49.68% (95%CI; 42.59–56.77; $I^2$ = 99.0%; p<0.001) respectively. The prevalence of under-weight among HIV positive children was found to be 49.67% in Ethiopia followed by 42.00 in Rwanda. It was high among cohort studies (44.87%). Based on the year of publication, the prevalence of under-weight among HIV positive children was found to be 40.88% from studies conducted from January 2008-December 2014, while it was 43.68% from studies conducted from 2015–2019. The prevalence of wasting among HIV positive children was found to be 29.7% in Tanzania followed by 24.94% in Ethiopia. Based on the study design, the prevalence of wasting among HIV positive children was found to be high in cohort studies (31.15%). The prevalence of stunting among HIV positive children was found to be 51.63% in Ethiopia, followed by 48.21% in Uganda.

## Conclusions

The results presented above provide evidence of a higher prevalence of under nutrition among HIV positive children in East Africa. Despite the country level variations of child under nutrition in East Africa, still it is high in all aspects compared to the studies from other parts of Africa. It is recommended that further systematic review and meta-analysis need to be conducted on magnitude of malnutrition among HIV positive children in Sub-Saharan Africa as a whole.

## Introduction

According to WHO malnutrition, in all its forms, includes under-nutrition (wasting, stunting, underweight), inadequate vitamins or minerals, overweight, obesity, and resulting diet-related non communicable diseases [1, 2]. The term under-nutrition is defined as the outcome of insufficient food intake (hunger) and repeated infectious disease; which includes being under-weight for once age, too short for once age(stunted), dangerously thin(wasted) and deficient in vitamins and minerals (micronutrient malnutrition) [3]. Malnutrition on the background of HIV infection is a complex medical condition that carries significant morbidity and mortality for affected children, with greater mortality from SAM among HIV-positive children than their HIV-negative peers [4].

Nutritional and micronutrient deficiencies are common in HIV-infected persons and play a major and synergistic role in disease progression and in the retardation of growth and development of children. HIV/AIDS is most prevalent in sub-Saharan Africa, where it combines with other common conditions such as malnutrition and opportunistic infections to cause devastation effect among families, communities, and nations [5, 6].

Human Immune Virus and under nutrition interact in a vicious cycle; HIV-induced immune impairment heightened risk of opportunistic infection and can worsen nutritional status of children. HIV infection often leads to nutritional deficiencies through decreased food intake, mal-absorption and increased utilization and excretion of nutrients, which in turn can hasten death. Likewise, nutritional status modulates the immunological response to HIV

infection, affecting the overall clinical outcome, since malnutrition is the primary cause of immunodeficiency [7–10].

Malnutrition is responsible for 11% of the total global disease burden and 35% of child deaths worldwide. In developing countries an estimated 19 million children are severely wasted [4, 11]. In some regions, notably sub-Saharan Africa, human immunodeficiency virus (HIV) infection poses an added challenge to the care of malnourished children. While the clinical context and interventions for many common causes of childhood mortality worldwide have been addressed over the last decade [12, 13], the management of malnutrition in children particularly in those infected with HIV remains poorly addressed [14].

In sub-Saharan Africa, the epidemiology of severe malnutrition has been increased in children requiring hospitalization composed of those who are HIV infected or HIV exposed with case-fatality rates reaches as high as 20–50% [15, 16]. Large percentages of HIV-positive children have an episode of severe malnutrition as their first AIDS-defining illness. Under nutrition is an important factor which might predict disease progression of HIV-infected individuals. It also results in higher risk of morbidity and mortality in both HIV-infected adults and children. Wasting and weight loss are common features of HIV infection, especially in resource-limited settings. It is known that children with HIV and severe malnutrition invariably have lower nutritional recovery and higher mortality rates than their HIV-negative counterparts [17, 18]. ART initiation in children can also cause metabolic disorders, and adverse effects on the nutritional status, especially in the first months of treatment, by causing complications, such as nausea and vomiting, or reduced bone mineral density [5, 19].

In East Africa, variety of previous studies have reported that the magnitude of under nutrition; accordingly the prevalence of under-weight, wasting and stunting was ranged from 19.4% to 77.1%, 7% to 77.1% and 13% to 71.8% respectively. This showed pronounced discrepancies among reports of under nutrition across different geographical settings and different time periods. Moreover, there is no regionally represented pooled data of under nutrition in East Africa. Therefore, this systematic review and meta-analysis was aimed to estimate the pooled prevalence of underweight, wasting and stunting in the East African context.

## Methods

### Review question

The review questions of this systematic review and meta-analysis were:

- What is the pooled prevalence of under-weight, wasting and stunting among HIV positive children in East Africa context?

### Study selection and screening

To exclude duplicate studies the retrieved studies were exported to Endnote version 8 reference managers. Before retrieval of full-text papers two investigators (BB and TG) independently screened the selected studies using article's title and abstracts. We used pre-specified inclusion criteria to further screen the full-text articles. Disagreements were resolved with third reviewer (GT) for the final decision on the selection of studies to be included in the analysis [20].

### Inclusion and exclusion criteria

Those studies had reported the prevalence of at least one under-weight, wasting and stunting and published in English language from January 2008 to December 2019. Studies conducted

with cross-sectional, and cohort study design was included. Studies conducted on marginalized groups/populations like children with any medical diseases, chronic diseases, or street mothers were excluded. Studies conducted on HIV/AIDS children in East Africa. The prevalence of under-weight, wasting, and stunting was considered when weight/ age, weight /height and height per age Z score <-2sd respectively within a specific population and multiply by 100 to be prevalence report.

## Searching strategy

This review identified studies that provide data on the prevalence of under-weight, wasting, and stunting with the context of Eastern Africa. In the searching engine, mainly PubMed, Cochrane library, Google Scholar, and Gray Literatures were retrieved. The last search date was December 30/2019. The Authors searched using keywords that are the amalgamations of population, condition/outcome, and context. A snowball searching of the references of relevant papers for linked articles were also performed. Those search terms or phrases including were: "children", "child", "infant", "under-nutrition", "underweight", "wasting", stunting, HIV, AIDS and Eastern Africa. Using those key terms, the following search map was applied: (prevalence OR magnitude) AND (children [MeSH Terms] OR child OR infant) AND (under-nutrition [MeSH Terms] OR underweight OR wasting OR stunting) AND (HIV OR AIDS) AND Eastern Africa on PubMed database (S1 Table). Thus, the PubMed search combines #1 AND #2 AND #3 (S1 Table). These search terms were further paired with names of each East African country. On both Cochran Library, and Google scholar, a build in text search were used on the advanced search section of the sources. The search date was December 30/ 2019.

## Quality assessment

The quality of the studies were appraised by three authors independently using the Joanna Briggs Institute (JBI) checklist [21]. The disagreement was resolved by the interference of a third reviewer. Studies were considered as high risk or poor quality, when it scored 3 and below [20–22] (S2 Table).

## Data extraction

The authors developed data extraction form on the excel sheet in considering country, year of publication, study design and prevalence of underweight, wasting and stunting reported. The data extraction sheet was piloted using selecting four papers randomly the data extraction sheet was piloted and adjustment were made after the template were piloted. Using the extraction form two authors (BB and TG) extracted the data in collaboration. The third author (GT) independently assessed the accuracy of the data and discussions with a third reviewer were done when disagreements between reviewers occurred [20, 22].

## Synthesis of results

The authors transformed the data to STATA 14 for analysis after it was extracted in excel sheet. Using a random effect meta-analysis model we pooled the overall prevalence estimates of underweight, wasting and stunting. Using Q statistic and the $I^2$ statistics we assessed the heterogeneity of effect size. The $I^2$ statistic value of zero, 25, 50, and 75% indicates true homogeneity, low heterogeneity, moderate heterogeneity and high heterogeneity, respectively. Using study design, study country, and year of publication subgroup analysis was done by. Sensitivity

analysis was done to assess the impact of a single study on the pooled estimate. Using funnel plot subjectively and objectively by Egger's regression test publication bias was assessed [20].

## Results

A total of 3094 studies were identified; 2050 from PubMed, 12 from Cochrane Library, 1010 from Google Scholar and 22 from other sources. After duplication removed, a total of 970 articles remained (2127 removed by duplication). Finally, 230 studies were screened for full-text review, and 22 articles with (n = 22074 patients) were selected for the analysis (Fig 1).

### Characteristics of included studies

Twenty two studies were included in this systematic review and meta-analysis [18, 25–45]. Of them 12 studies were done in Ethiopia [18, 25–34, 37], 1 in Kenya [41], while 2 were in Uganda

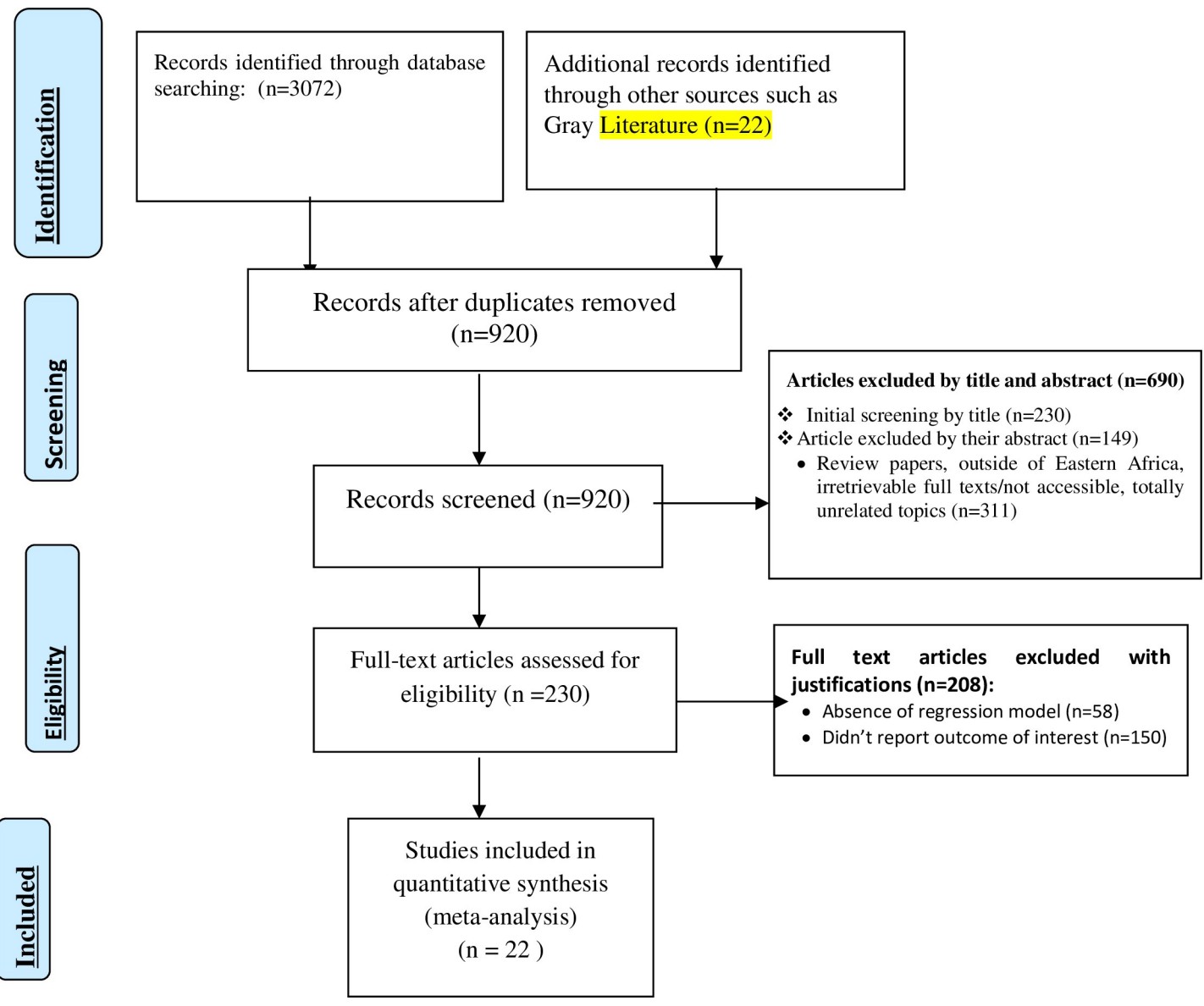

**Fig 1. PRISMA–Adapted flow diagram showed the results of the search and reasons for exclusion [23, 24].**

[38, 39], 1 in Rwanda [35], and 6 in Tanzania [36, 40, 42–45]. Based on the study design used 14 studies were done by cross-sectional study design [18, 26, 27, 30, 31, 33–35, 37, 38, 40, 41, 43, 45] and while other 8 studies were conducted by cohort study design [25, 27–29, 32, 36, 39, 42, 44]. 14/22(63.6%) were published between 2008 and 2014 and the remaining 8/22 (33.4%) were published between 2015 and 2019. The total number of participants in the included studies were ranges from 96 [32] to 5951 [38] (Table 1).

## Meta-analysis

**Underweight.** *Prevalence of under-weight among HIV positive children.* Most of the studies (n = 17) have reported the prevalence of under-weight among HIV positive children. The prevalence of under-weight was ranged from 19.4% [41] to 77.1% [32]. The pooled prevalence of under-weight among HIV positive children in East Africa using random-effects model analysis was found to be 41.63% (95%CI; 35.69–47.57; $I^2$ = 98.7%; p<0.001) (Fig 2).

*Subgroup analysis for the prevalence of under-weight among HIV positive children in Eastern Africa.* Using country, study design, and year of publication as criteria subgroup analysis was done. Based on this, the prevalence of under-weight among HIV positive children was found to be 49.67% in Ethiopia, 42.00 in Rwanda, 38.59% in Tanzania, 34.52% in Uganda, and 19.4% in Kenya (Fig 3 and Table 2). The prevalence of under-weight among HIV positive children was found to be 39.33% in cross-sectional studies and 44.87% in cohort studies (Fig 4 and Table 2). It was found to be 40.88% from studies conducted from January 2008-December 2014, while it was 43.68% from studies conducted from 2015–2019 (Fig 5 and Table 2).

**Sensitivity analysis for under-weight.** To assess the influence of individual study on the pooled prevalence of under-weight in Eastern Africa we employed a sensitivity analysis.

**Table 1. Distribution of included studies on the prevalence underweight, wasting and stunting in East Africa, from January 2008-December 2019.**

| Author name | Publication Year | Country | Region | Study design | Sample size | Under weight | Wasting | Stunting | Cut of point Z-scor | Reference |
|---|---|---|---|---|---|---|---|---|---|---|
| 1. Kedir *et al* | 2014 | Ethiopia | Adama/Et | Cohort | 560 | 51.6 | | | <-2sd | [25] |
| 2. Mekonen *et al* | 2014 | Ethiopia | AA/ET | Cross sectional | 255 | 47.5 | | 71.3 | <-2sd | [26] |
| 3. Jeylan *et al* | 2018 | Ethiopia | Adama/ET | Cross sectional | 412 | | 21.8 | 13.4 | <-2sd | [27] |
| 4. Tekleab *et al* | 2016 | Ethiopia | AA/ET | Cohort | 202 | 39.5 | 16.3 | 71.3 | <-2sd | [28] |
| 5. Yassin *et al* | 2017 | Ethiopia | Fiche/ET | Cohort | 269 | | | | <3rd centile | [29] |
| 6. Abdulkadir *et al* | 2014 | Ethiopia | Gonder/ET | Cross sectional | 142 | | 31.7 | 46.5 | NR | [30] |
| 7. Haileselassie *et al* | 2019 | Ethiopia | Harar/Et | Cross sectional | 390 | | 28.2 | 24.7 | <-2sd | [31] |
| 8. Teklemariam *et al* | 2015 | Ethiopia | Harar/ET | Cross sectional | 108 | 51.6 | 31.5 | 49.1 | <-2sd | [18] |
| 9. Workneh *et al* | 2009 | Ethiopia | Jimma/ET | Cohort | 96 | 77.1 | 47.5 | 63.5 | <5th centile | [32] |
| 10. Wondimu *et al* | 2014 | Ethiopia | Hawassa/Et | Cross sectional | 455 | 41.2 | 21.4 | 60.5 | <-2sd | [33] |
| 11. Megabiaw *et al* | 2012 | Ethiopia | Gondar/ET | Cross sectional | 301 | 41.7 | 5.8 | 65 | <-2sd | [34] |
| 12. Arpadi *et al* | 2019 | Rwanda | Rwanda | Cross sectional | 374 | 42 | | | <-2sd | [35] |
| 13. Kamenju *et al* | 2017 | Tanzania | Tanzania | Cohort | 2092 | 25.4 | 21.6 | 27.1 | <-2sd | [36] |
| 14. **Sewale** *et al* | 2018 | Ethiopia | Gojjam/Et | Cross sectional | 372 | | | | <-2 sd | [37] |
| 15. Nalwoga *et al* | 2010 | Uganda | Uganda | Cross sectional | 5951 | 30 | 10 | 42 | <-2 sd | [38] |
| 16. Arinaitwe *et al* | 2012 | Uganda | Uganda | Cohort | 358 | 39.7 | | 71.8 | <-2 sd m-s | [39] |
| 17. Sunguya *et al* | 2014 | Tanzania | Tanzania | Cross sectional | 748 | 40.6 | 30.2 | 60.8 | ≥-2sd | [40] |
| 18. Herman *et al* | 2102 | Kenya | Kenya | Cross sectional | 2275 | 19.4 | 7 | 28.6 | <-2 sd | [41] |
| 19. Mwiru *et al* | 2015 | Tanzania | Tanzania | Cohort | 3144 | 53 | 33 | 56 | <-2 sd | [42] |
| 20. Sunguya *et al* | 2011 | Tanzania | Tanzania | Cross sectional | 213 | 22.1 | 23.6 | 36.6 | <-2sd | [43] |
| 21. Mwiru *et al* | 2014 | Tanzania | Tanzania | Cohort | 3144 | 30 | 40 | 52 | <-2sd | [44] |
| 22. Dundigalla *et al* | 2015 | Tanzania | Tanzania | Cross sectional | 213 | 61 | 29.1 | 56.8 | <-2sd | [45] |

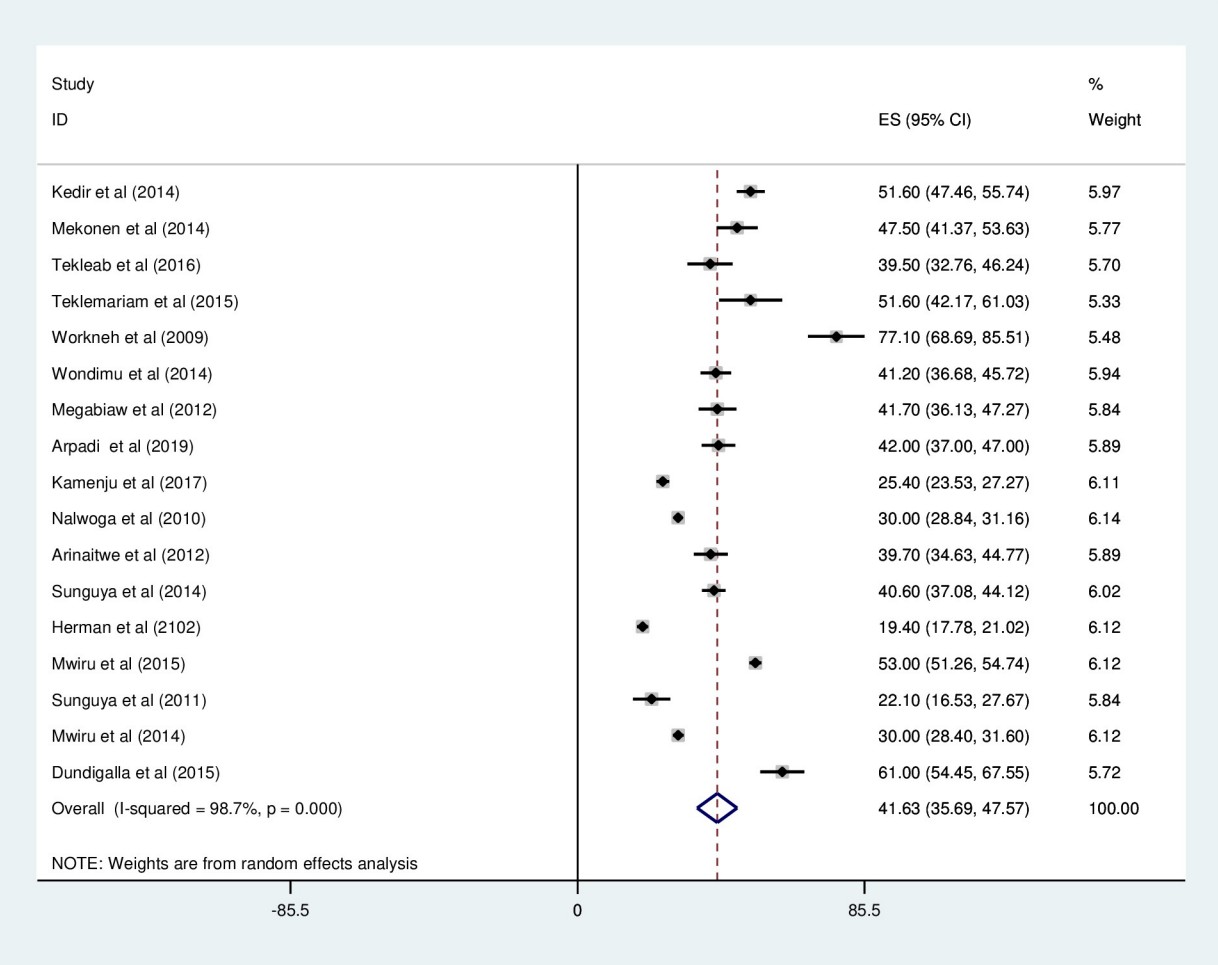

**Fig 2. Forest plot showing the pooled prevalence of under-weight among HIV positive children in East Africa, from January 2008-December 2019.**

Accordingly the findings were not dependent on all included studies. The prevalence of under-weight varied between 39.55(33.67–45.43) [32] and 42.71(36.31–49.11) [36] after deletion of a single study (S1 Fig).

**Publication bias.** Subjectively the funnel plot indicated symmetrical distributions which indicate the absence of publication bias. The Egger's regression test- also revealed the absence of publication bias; P-*value* was 0.064, (S2 Fig).

**Wasting.** *Prevalence of wasting among HIV positive children in East Africa.* Most of the studies (n = 16) have reported the prevalence of wasting among HIV positive children. The prevalence of wasting was ranged from 7% [41] to 77.1% [32]. The prevalence of wasting among HIV positive children in East Africa was found to be 24.65% (95%CI; 18.34–30.95; $I^2$ = 99.2%; p<0.001) (Fig 6).

*Subgroup analysis of the prevalence of wasting among HIV positive children in Eastern Africa.* The subgroup analysis was by country, study design, and year of publication. Accordingly, the prevalence of wasting among HIV positive children was found to be 24.94% in Ethiopia, 29.7% in Tanzania, 10.0% in Uganda, and 7.0% in Kenya (Fig 7 and Table 3). The prevalence of wasting among HIV positive children was found to be 21.22% in cross-sectional studies and

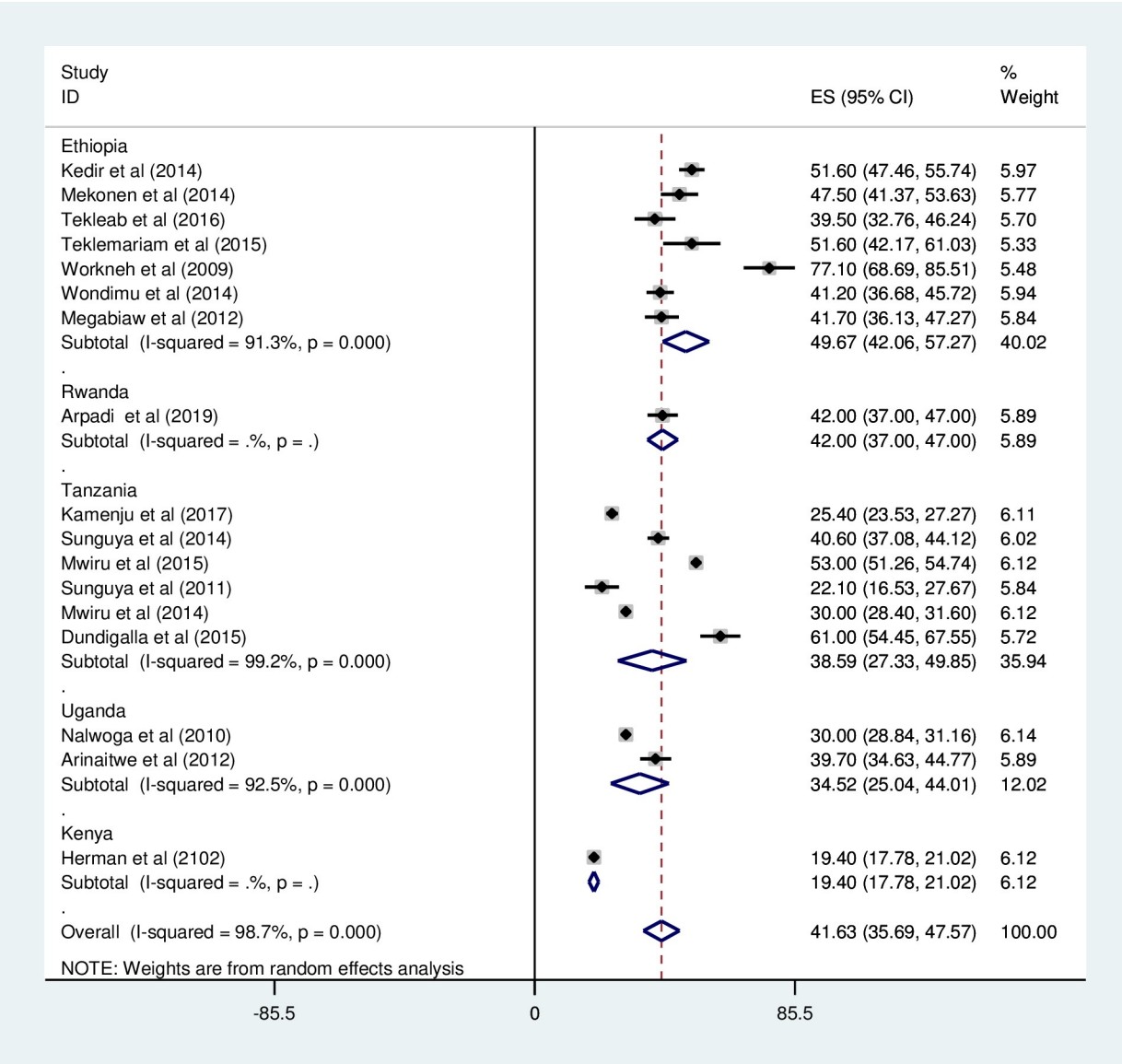

**Fig 3. Forest plot showing the subgroup analysis of the prevalence of under-weight among HIV positive children by country in East Africa, from January 2008-December 2019.**

31.15% in cohort studies (Fig 8 and Table 3). Based on the year of publication, the prevalence of wasting among HIV positive children was found to be 24.7% from studies conducted from January 2008-December 2014, while it was 24.02% from studies conducted from 2015–2019 (Fig 9 and Table 3).

**Sensitivity analysis for wasting.** To assess the influence of individual study on the pooled prevalence of wasting in Eastern Africa we employed a sensitivity analysis. Accordingly the findings were not dependent on all included studies. The prevalence of wasting varied between 23.29(16.85–29.74) [32] and 25.95(19.34–32.52) [34] after deletion of a single study (S3 Fig).

**Publication bias.** Subjectively the funnel plot indicated symmetrical distributions which indicate the absence of publication bias. The Egger's regression test- also revealed the absence of publication bias; P-*value* was 0.068 (S4 Fig).

**Table 2. Summery of subgroup analysis of the prevalence of under-weight among HIV positive children in Eastern Africa by country, design and year of publication, from January 2008-December 2019.**

| Variables | Characteristics | Pooled prevalence, %(95% CI) | I², (P-value) |
|---|---|---|---|
| By country | Ethiopia | 49.67(42.06–57.27) | 91.3%(<0.001) |
| | Rwanda | 42.00(37.00–47.00) | - |
| | Tanzania | 38.59(27.33–49.83) | 99.2%(<0.001) |
| | Uganda | 34.52(25.04–44.01) | 92.5%(<0.001) |
| | Kenya | 19.40(17.78–21.02) | - |
| By study design | Cross-sectional | 39.33(32.54–46.12) | 97.8% (<0.001) |
| | Cohort | 44.87(33.97–55.77) | 99.1%(<0.001) |
| By year of publication | 2008–2014 | 40.88(33.58–48.17) | 99.0% (<0.001) |
| | 2015–2019 | 43.68(29.57–57.78) | 97.5%(<0.001) |

**Stunting.** *Prevalence of stunting among HIV positive children.* Most of the studies (n = 18) have reported the prevalence of stunting among HIV positive children. The prevalence of stunting was ranged from 13% [27] to 71.8% [39]. The pooled prevalence of stunting among

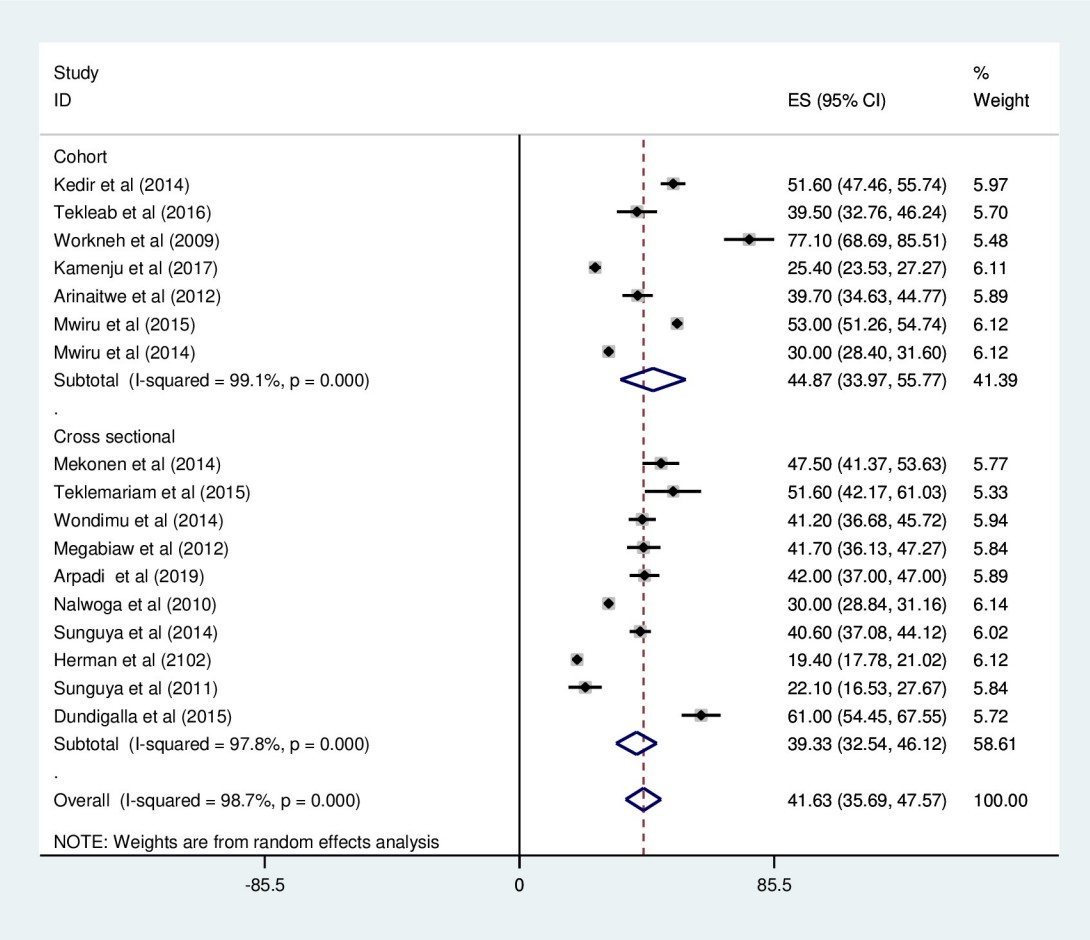

**Fig 4. Forest plot showing the subgroup analysis of the prevalence of under-weight among HIV positive children by study design in East Africa, from January 2008-December 2019.**

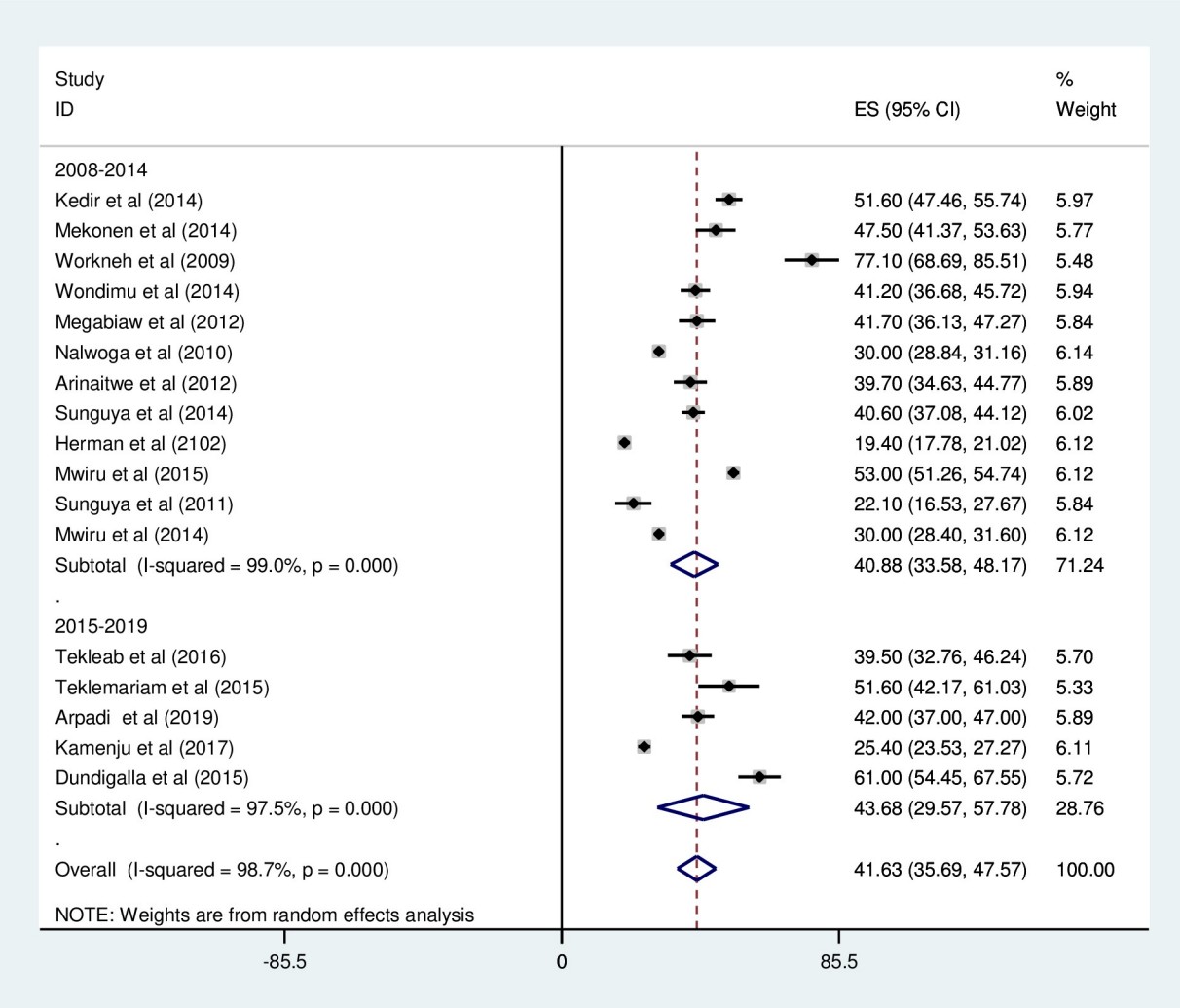

**Fig 5. Forest plot showing the subgroup analysis of the prevalence of under-weight among HIV positive children by year of publication in East Africa, from January 2008-December 2019.**

HIV positive children in East Africa using a random-effects model analysis was found to be 49.68% (95%CI; 42.59–56.77; $I^2$ = 99.0%; p<0.001) (Fig 10).

*Subgroup analysis of the prevalence of stunting among HIV positive children in Eastern Africa.* Using country, study design, and year of publication as criteria subgroup analysis was done. Accordingly, the prevalence of stunting among HIV positive children was found to be 51.63% in Ethiopia, 38.59% in Tanzania, 48.21% in Uganda, and 28.60% in Kenya (Fig 11 and Table 4). Based on the study design, the prevalence of under-weight among HIV positive children was found to be 46.16% in cross-sectional studies and 56.73% in cohort studies (Fig 12 and Table 4). The prevalence of stunting among HIV positive children was found to be 54.44% from studies conducted from January 2008-December 2014, while it was 40.11% from studies conducted from 2015–2019 (Fig 13 and Table 4).

**Sensitivity analysis.** To assess the influence of individual study on the pooled prevalence of stunting in Eastern Africa we employed a sensitivity analysis. Accordingly the findings were

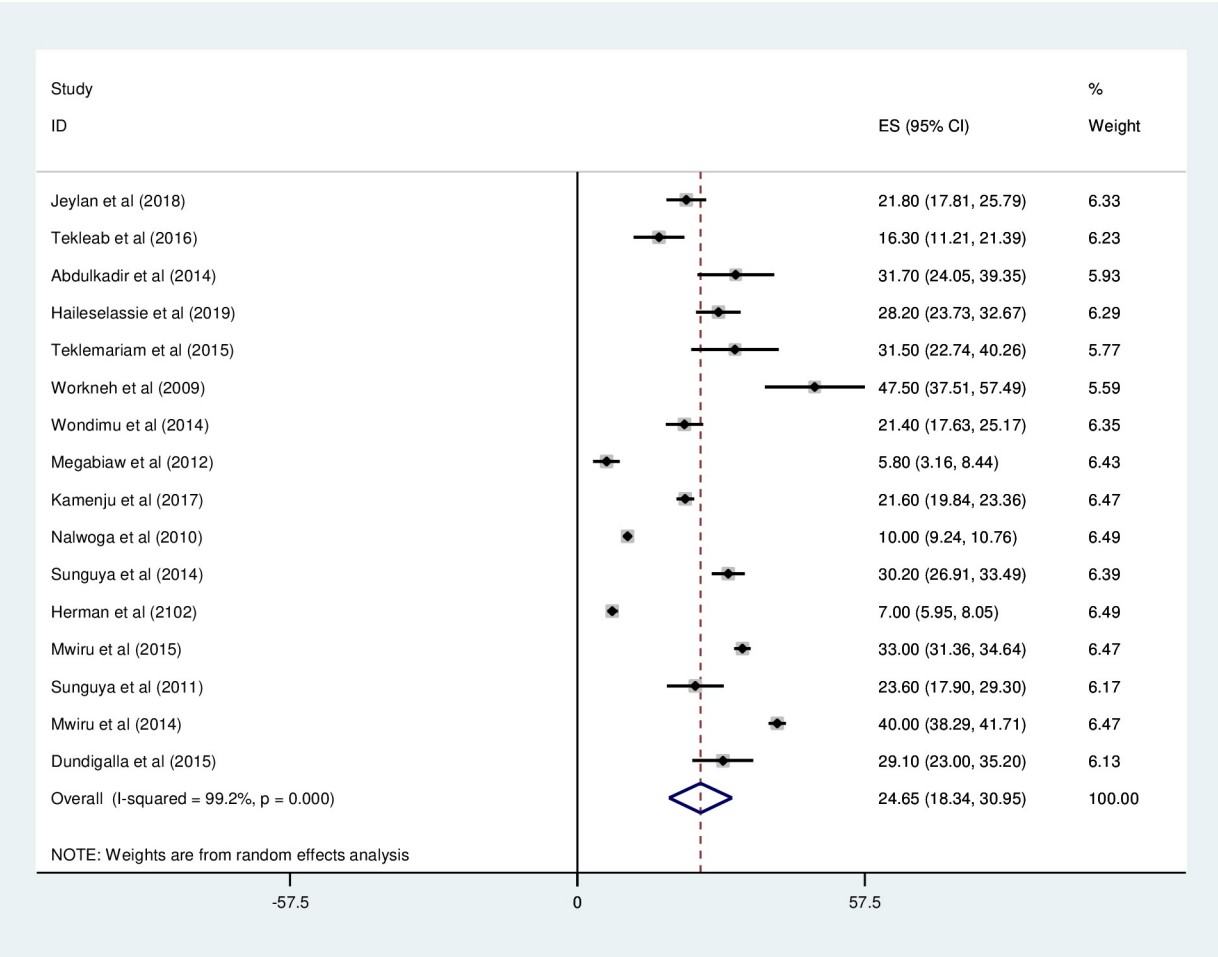

**Fig 6. Forest plot shows the pooled prevalence of wasting among HIV positive children in East Africa, from January 2008-December 2019.**

not dependent on all included studies. The prevalence of stunting varied between 48.3(41.27–55.44) [39] and 51.8(45.07–58.62) [27] after deletion of a single study (S5 Fig).

**Publication bias.** Subjectively the funnel plot indicated symmetrical distributions which indicate the absence of publication bias. The Egger's regression test- also revealed the absence of publication bias; P-*value* was 0.068 (S6 Fig).

## Discussion

Based on this systematic review and meta-analysis, it was found that the pooled prevalence of underweight in eastern Africa is 41%. This result is higher than the study conducted among HIV positive children, in Nigeria (12.1%), Cameroon (20.5%, 37.8%) and, and Burkina Faso (31%) [46–49] respectively. But this result is lower compared to large scale study conducted in southern Africa(47.3%) [50]. The discrepancy might be due to the difference in number of study participants across the studies.

The sub group analysis based on country revealed that, the pooled prevalence of underweight among HIV positive children was found to be 49.67% in Ethiopia, followed by 42.00% in Rwanda. The result is higher compared to large scale DHS study in sub Saharan Africa,

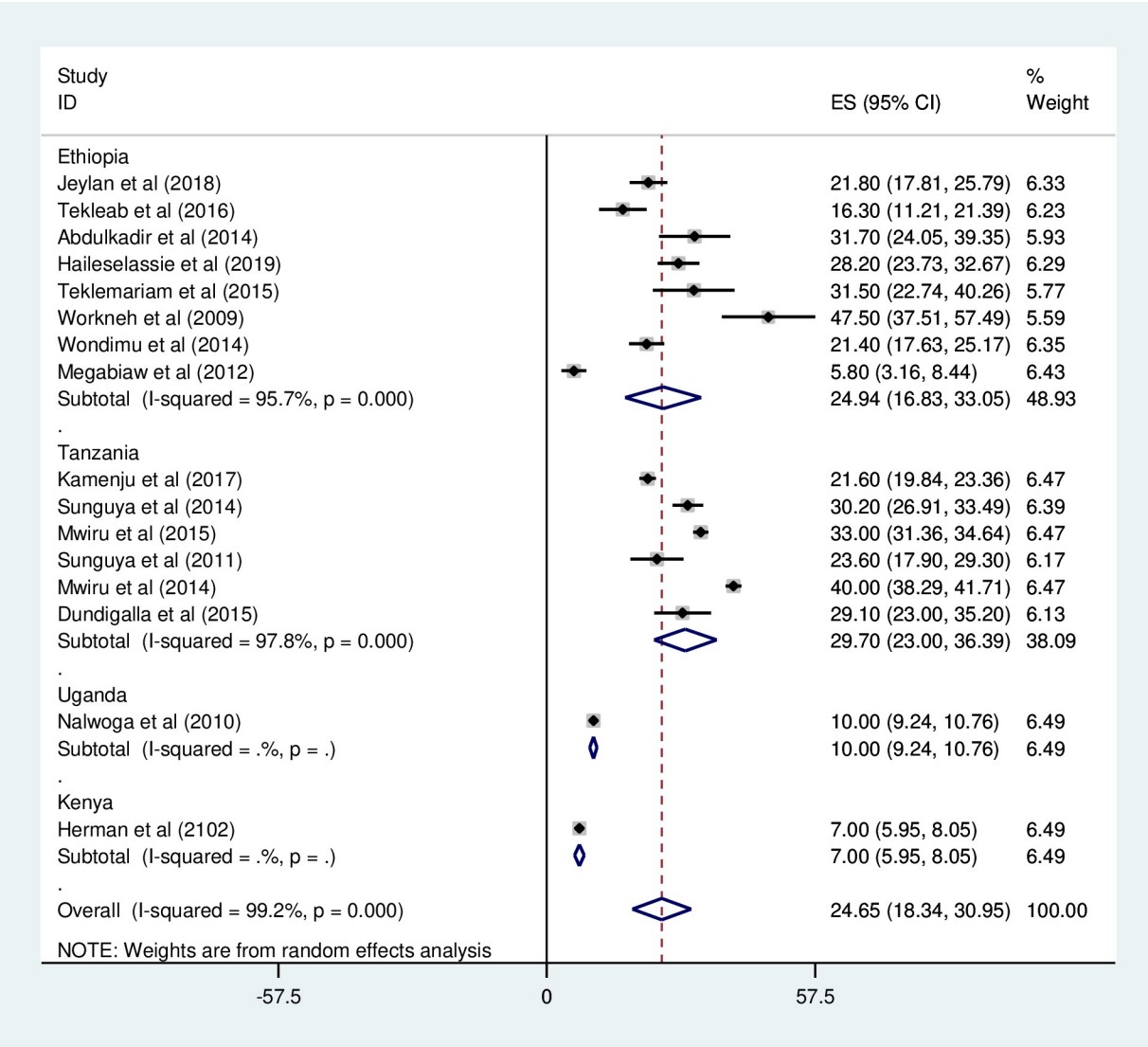

**Fig 7. Forest plot shows the pooled prevalence of wasting by country in East Africa, from January 2008-December 2019.**

**Table 3. Summery of subgroup analysis of the prevalence of wasting among HIV positive children in Eastern Africa by country, design and year of publication, from January 2008-December 2019.**

| Variables | Characteristics | Pooled prevalence, %(95% CI) | $I^2$, (P-value) |
|---|---|---|---|
| By country | Ethiopia | 24.94(16.83–33.05) | 95.7%(<0.001) |
| | Tanzania | 29.7(23.00–36.39) | 97.8%(<0.001) |
| | Uganda | 10.0(9.24–10.76) | - |
| | Kenya | 7.0(5.95–8.95) | - |
| By study design | Cross-sectional | 21.22(16.56–25.88) | 97.6% (<0.001) |
| | Cohort | 31.15(22.57–39.74) | 98.5%(<0.001) |
| By year of publication | 2008–2014 | 24.7(16.11–33.30) | 99.5% (<0.001) |
| | 2015–2019 | 24.02(20.35–27.70) | 76.9%(<0.001) |

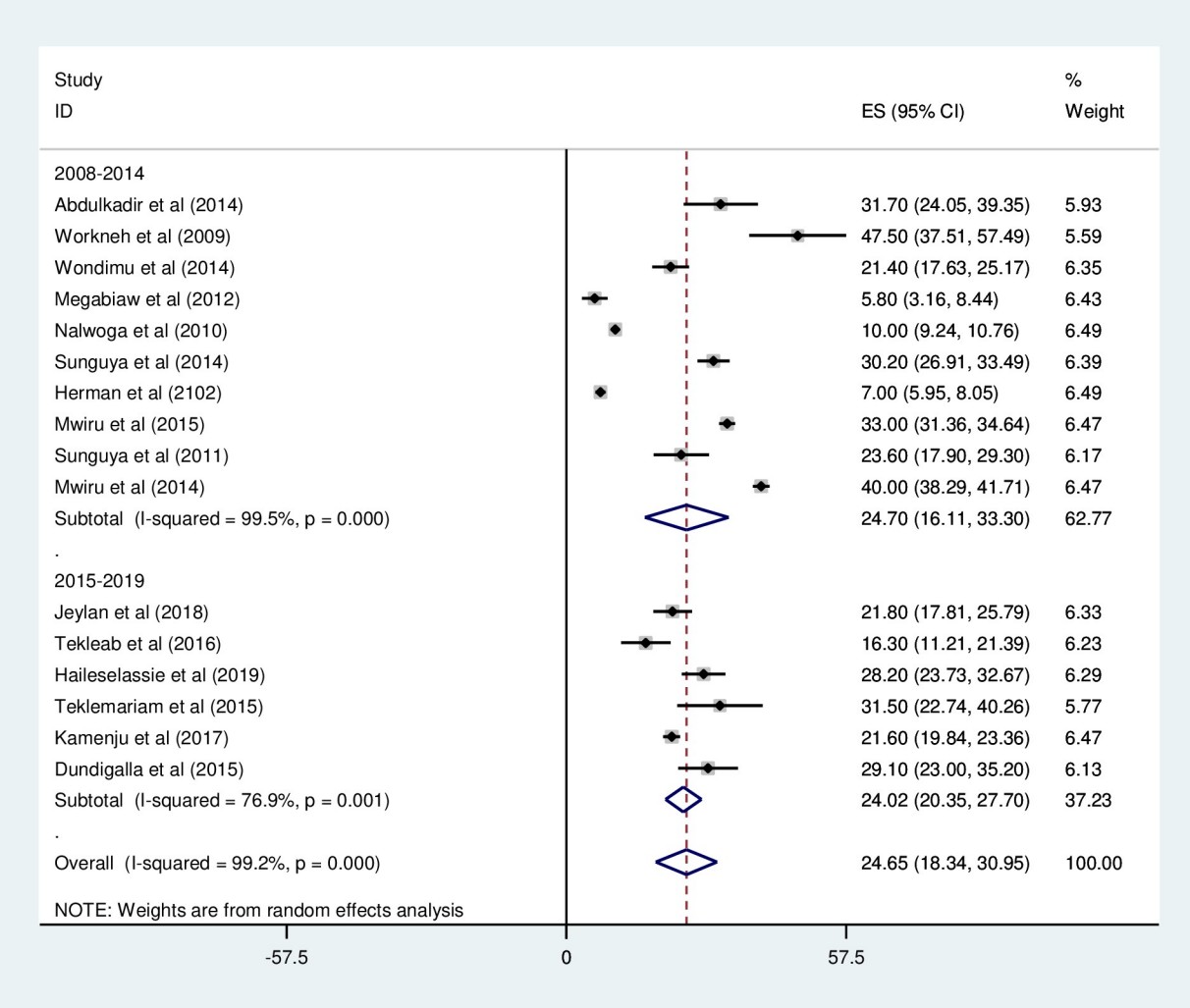

**Fig 8. Forest plot shows the pooled prevalence of wasting by year of publication in East Africa, from January 2008-December 2019.**

which accounts 31.2% in Ethiopia and 18.5% in Rwanda respectively [51], and EDHS 2109 mini report which accounts overall 21% [52]. Based on the study design, the pooled prevalence of under-weight among HIV positive children was 39.33% in cross-sectional studies and 44.87% in cohort studies respectively. This may be due to cohort studies apply strict follow-up trend of the patients; through this they can record more reliable reports of the patients overall character. The pooled prevalence of underweight on the studies conducted from (2015–2019) found to be increased (43.68%) compared to the studies conducted from January 2008-2014 (40.88%). This indicates that underweight is still an alarming issue among HIV positive children's in East Africa.

The pooled prevalence of wasting among HIV positive children in East Africa found to be 24.65% (95%CI; 18.34–30.95). This result is higher compared to the study conducted in central and West African countries (16%) [9], the study conducted by Pendal et al in Cameroon (18.4%) [49], and large scale study conducted in southern Africa(21.3%) [50]. The discrepancy might be due to the emphasis given by the government as well as stakeholders of the area regarding the effects of HIV/AIDS on child growth and development.

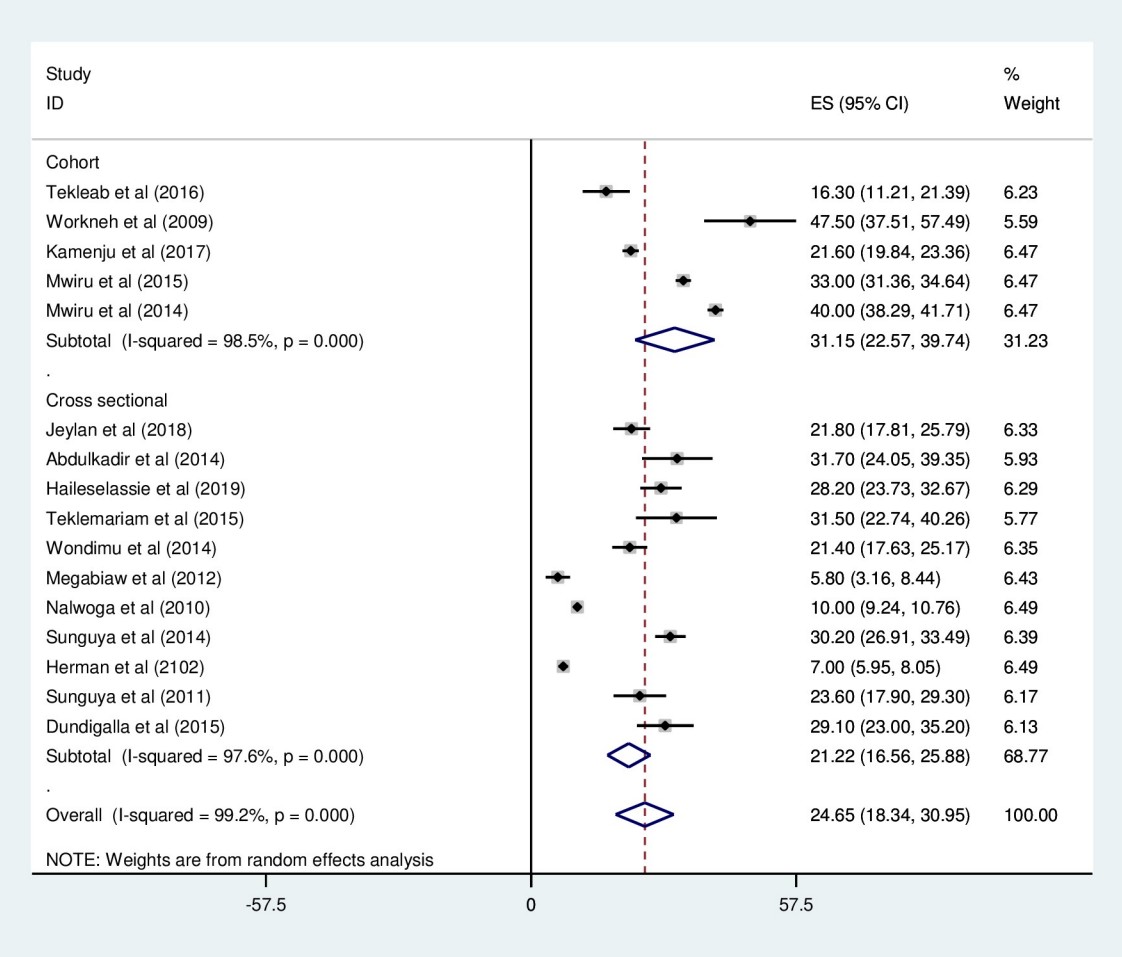

**Fig 9. Forest plot shows the pooled prevalence of wasting by study design in East Africa, from January 2008-December 2019.**

Regarding sub group analysis by country the prevalence of wasting is higher in Tanzania (29%) followed by Ethiopia (24.94%). This result is higher compared to the data reported by Ethiopian Demographic Health Survey (EDHS), 2019 mini report which accounted 7% of overall children wasted [52]. This shows that these two countries needs great emphasis to decrease the burden of acute under-nutrition due to HIV infection; and it is better to invite governmental and non-governmental organizations regarding nutritional support to HIV-infected children at ART initiation. The prevalence of wasting among HIV positive children in studies conducted by follow up were found to be higher compared to cross-sectional once.

The pooled prevalence of stunting among HIV positive children in East Africa found to be 49.68% (95%CI; 42.59–56.77). This result is higher compared to large scale study conducted on children with HIV positive in Central and West African (33%) HIV care programmes supported by the Growing up Programme in 2011 [9]. But this study is lower than the study conducted in southern Africa, which accounts (61.1%) of HIV positive children were chronically under nourished. This difference might be due to the action of non-governmental organizations on providing child nutrition compared to our study area.

The sub group analysis result based on country shows, greater than half (51.63%) of HIV positive children's in Ethiopia found to be stunted followed by nearly half (48.21%) of HIV

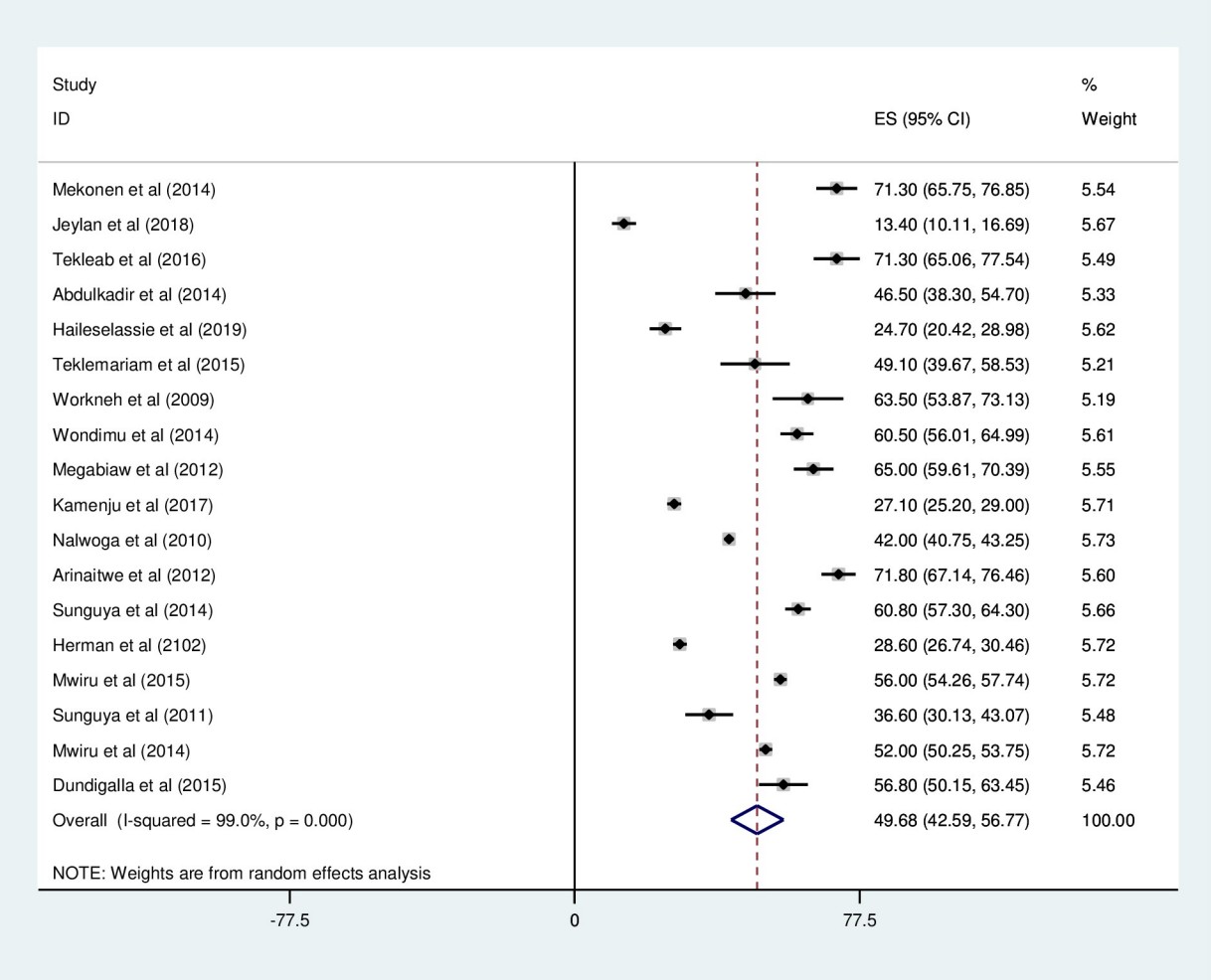

**Fig 10. Forest plot shows the pooled prevalence of stunting in East Africa, from January 2008-December 2019.**

positive children's in Uganda are under this scheme. This result is higher compared the data reported by mini EDHS, 2019 accounted (37%) [52], and large scale DHS study conducted in sub-Saharan countries, which accounted 26.2% in Ethiopia and 38.2% in Rwanda [51]. The inconsistency between results might be due to difference number and geographical area of study participants. This result calls the integration of nutritional support for HIV positive children and early initiation of ART to loosen the burden of chronic under nutrition in East African Countries.

The relationship between nutrition and HIV infection is very complex and is modified by factors such as nutritional status, including wasting or obesity, and micronutrient deficiencies along with HIV disease stage. Starting assessment, counseling, and education regarding nutrition shortly after HIV diagnosis is imperative. Good nutrition has been proven to increase resistance to infection and disease and improves energy. Severe malnutrition in HIV-infected persons is recognized as the "wasting syndrome," defined by the Centers for Disease Control and Prevention (CDC) as a body weight loss equal to or greater than 10% with associated fatigue, fever, and diarrhea unexplained by another cause [53].

HIV positive children are the most vulnerable group for underweight, wasting and stunting among and need more medical and research attention [54].

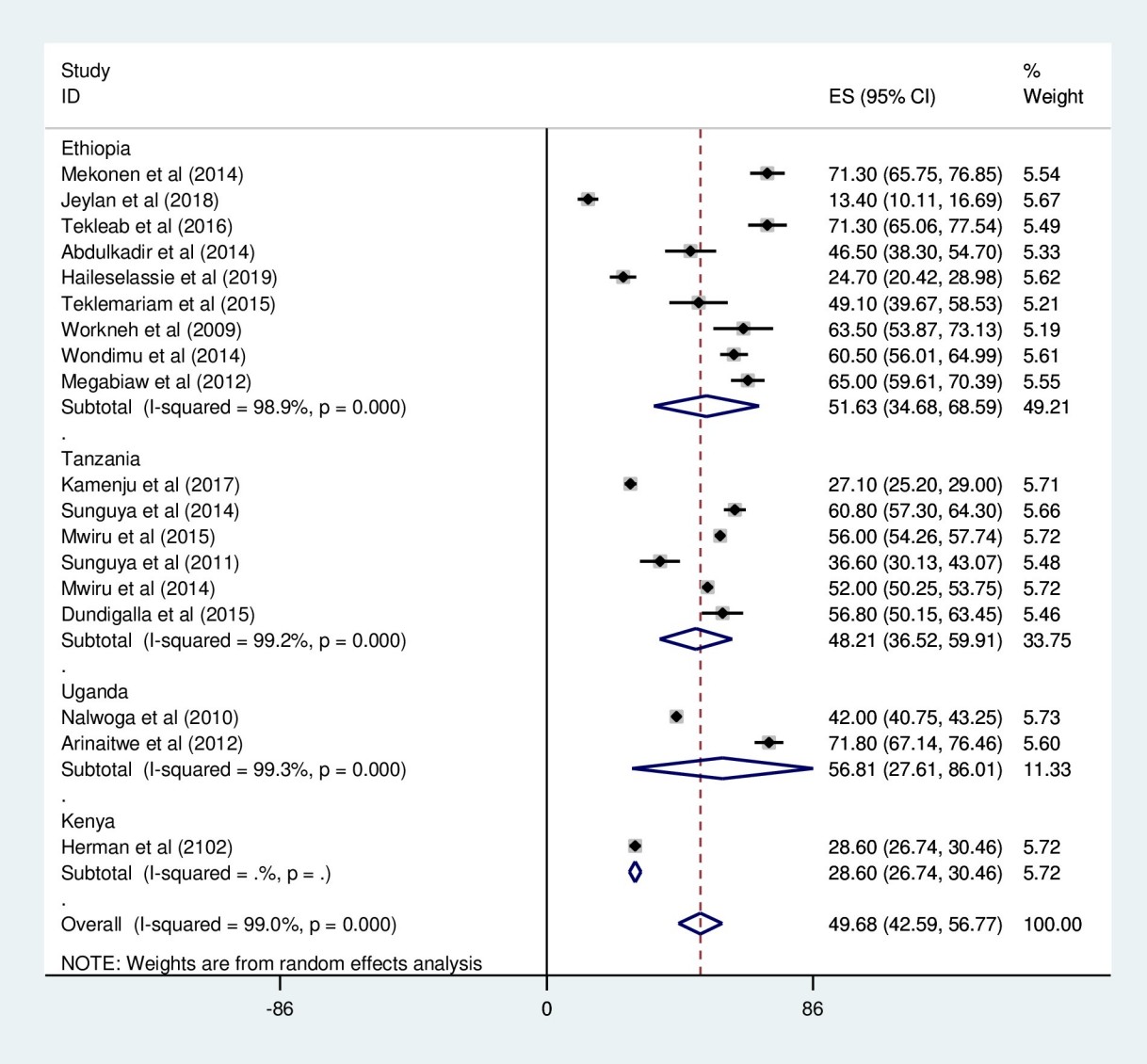

**Fig 11. Forest plot shows the pooled prevalence of stunting by country in East Africa, from January 2008-December 2019.**

**Table 4. Summery of subgroup analysis of the prevalence of stunting among HIV positive children in Eastern Africa by country, design and year of publication, from January 2008-December 2019.**

| Variables | Characteristics | Pooled prevalence, %(95% CI) | $I^2$, (P-value) |
|---|---|---|---|
| By country | Ethiopia | 51.63(34.68–68.59) | 98.9%(<0.001) |
| | Tanzania | 48.21(36.52–59.91) | 99.2%(<0.001) |
| | Uganda | 56.81(27.61–86.02) | 99.3%(<0.001) |
| | Kenya | 28.60(26.74–30.46) | - |
| By study design | Cross-sectional | 46.16(37.21–55.11) | 98.8% (<0.001) |
| | Cohort | 56.73(43.68–69.77) | 99.3%(<0.001) |
| By year of publication | 2008–2014 | 54.44(47.20–61.68) | 98.8% (<0.001) |
| | 2015–2019 | 40.11(25.93–54.30) | 98.6%(<0.001) |

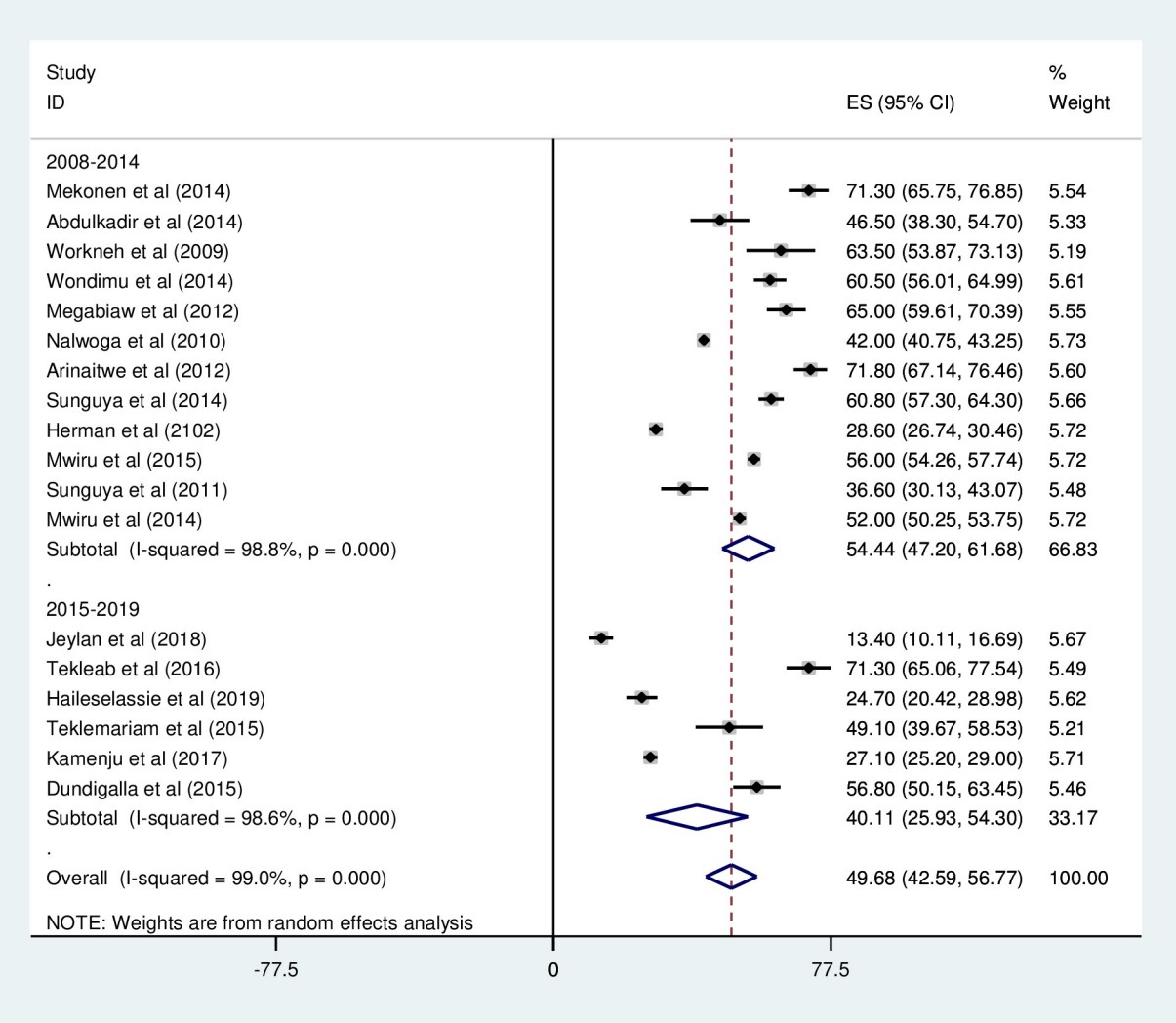

| Study ID | ES (95% CI) | % Weight |
|---|---|---|
| **2008-2014** | | |
| Mekonen et al (2014) | 71.30 (65.75, 76.85) | 5.54 |
| Abdulkadir et al (2014) | 46.50 (38.30, 54.70) | 5.33 |
| Workneh et al (2009) | 63.50 (53.87, 73.13) | 5.19 |
| Wondimu et al (2014) | 60.50 (56.01, 64.99) | 5.61 |
| Megabiaw et al (2012) | 65.00 (59.61, 70.39) | 5.55 |
| Nalwoga et al (2010) | 42.00 (40.75, 43.25) | 5.73 |
| Arinaitwe et al (2012) | 71.80 (67.14, 76.46) | 5.60 |
| Sunguya et al (2014) | 60.80 (57.30, 64.30) | 5.66 |
| Herman et al (2102) | 28.60 (26.74, 30.46) | 5.72 |
| Mwiru et al (2015) | 56.00 (54.26, 57.74) | 5.72 |
| Sunguya et al (2011) | 36.60 (30.13, 43.07) | 5.48 |
| Mwiru et al (2014) | 52.00 (50.25, 53.75) | 5.72 |
| Subtotal (I-squared = 98.8%, p = 0.000) | 54.44 (47.20, 61.68) | 66.83 |
| . | | |
| **2015-2019** | | |
| Jeylan et al (2018) | 13.40 (10.11, 16.69) | 5.67 |
| Tekleab et al (2016) | 71.30 (65.06, 77.54) | 5.49 |
| Haileselassie et al (2019) | 24.70 (20.42, 28.98) | 5.62 |
| Teklemariam et al (2015) | 49.10 (39.67, 58.53) | 5.21 |
| Kamenju et al (2017) | 27.10 (25.20, 29.00) | 5.71 |
| Dundigalla et al (2015) | 56.80 (50.15, 63.45) | 5.46 |
| Subtotal (I-squared = 98.6%, p = 0.000) | 40.11 (25.93, 54.30) | 33.17 |
| . | | |
| Overall (I-squared = 99.0%, p = 0.000) | 49.68 (42.59, 56.77) | 100.00 |

NOTE: Weights are from random effects analysis

-77.5    0    77.5

**Fig 12. Forest plot shows the pooled prevalence of stunting by year of publication in East Africa, from January 2008-December 2019.**

## Conclusion

The findings of this review results revealed a higher prevalence of under-nutrition among HIV positive children in East Africa. Despite the country level variations of child under-nutrition in East Africa, still it is high in all aspects compared to the studies from other parts of Africa. It is recommended that further systematic review and meta-analysis need to be conducted on magnitude of malnutrition among HIV positive children in Sub-Saharan Africa as a whole.

## Strength and limitations

As strength the authors used a standardized JBI quality assessment checklist and the included studies were low risk of bias. Moreover, we employed subgroup analysis based on study country, study design, and year of publication and sensitivity analysis to identify the small study effect and the risk of heterogeneity. Nevertheless, there are a few limitations to consider in the present study. First, due to the cross-sectional design, the observed results cannot be interpreted as causal. Second, the self-reported measures of variables are subject to measurement,

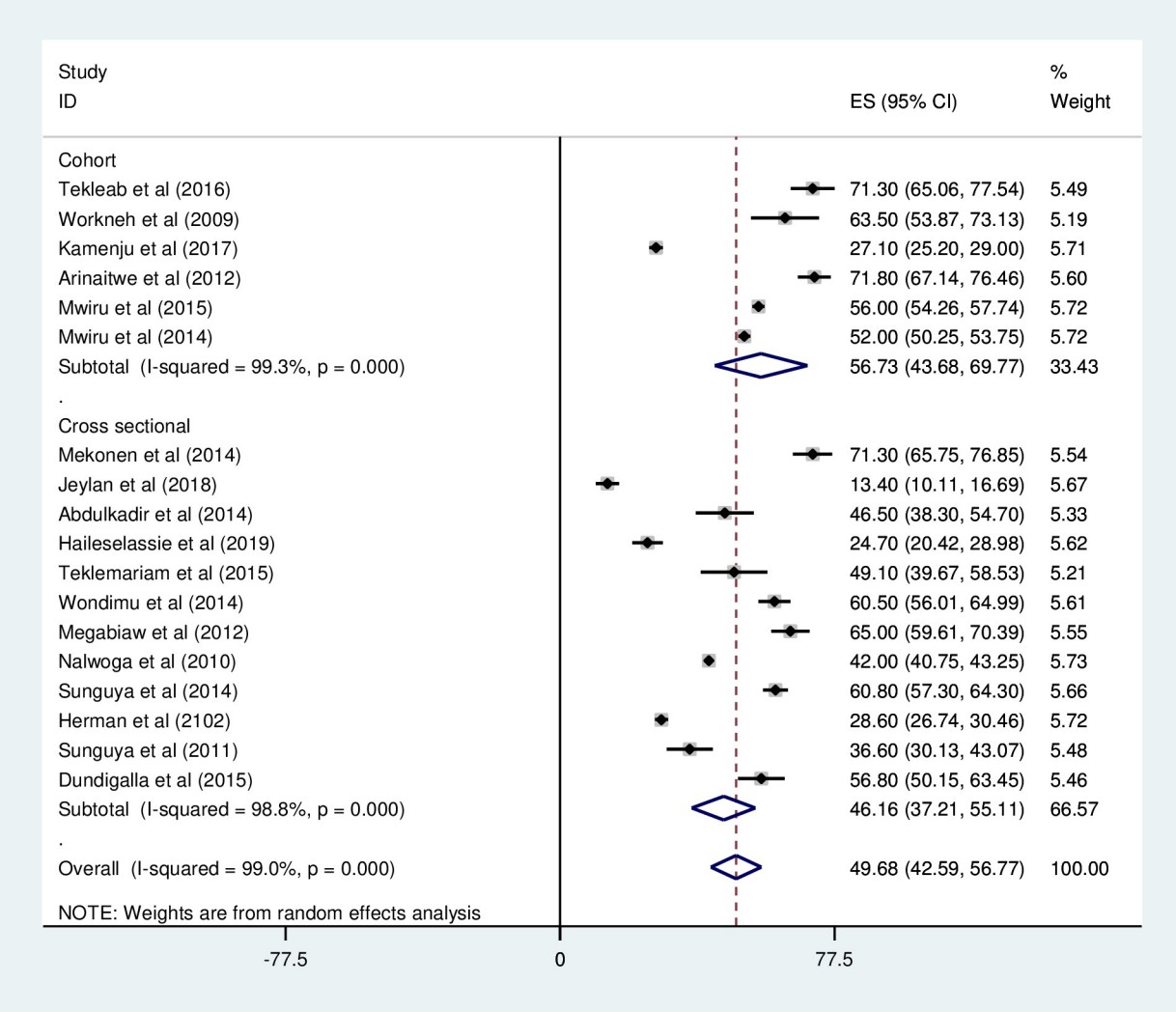

**Fig 13. Forest plot shows the pooled prevalence of stunting by study design in East Africa, from January 2008-December 2019.**

self-report, social desirability, and recall biases. Third, publication bias may occur because all grey literature may not be included and language bias; since all included studies are published in English only. Forth, since only HIV positive children were taken as a study subjects it is difficult to present the result comparing with HIV negative children.

## Recommendations

Special attention and efforts to reduce the burden of under-nutrion in HIV positive children should be applied in East Africa. Health professionals working with HIV positive children should routinely screen and manage under nutrition (underweight, wasting and stunting). Policy makers should incorporate strategies regarding prevention, screening and management of under nutrition in management of HIV/AIDS. Parents of HIV positive children should improve their feeding practice so as to prevent under nutrition.

Hence, nutritional attention is needed for children living with HIV/AIDS and at the time of ART initiation. Future researchers should conduct comparative studies on nutritional status

between HIV positive and negative children so that meta-analysis of those comparative studies is recommended by authors of this study.

## Supporting information

**S1 Checklist.**
(DOC)

**S1 Table. Search strategy used for one of the databases.**
(DOCX)

**S2 Table. Quality appraisal result of included studies in East Africa, from January 2000-December 2019.** Using Joanna Briggs Institute (JBI) quality appraisal checklist [16].
(DOCX)

**S1 Fig. Forest plot showing the sensitivity analysis of the prevalence of under-weight among HIV positive children in East Africa, from January 2008-December 2019.**
(DOCX)

**S2 Fig. Publication bias of the prevalence of under-weight among HIV positive children in East Africa, from January 2008-December 2019.**
(DOCX)

**S3 Fig. Sensitivity analysis of the pooled prevalence of wasting in East Africa, from January 2008-December 2019.**
(DOCX)

**S4 Fig. Publication bias of the pooled prevalence of wasting in East Africa, from January 2008-December 2019.**
(DOCX)

**S5 Fig. Sensitivity analysis for the pooled prevalence of stunting in East Africa, from January 2008-December 2019.**
(DOCX)

**S6 Fig. Publication bias the pooled prevalence of stunting in East Africa, from January 2008-December 2019.**
(DOCX)

**S1 Dataset.**
(DOCX)

## Author Contributions

**Conceptualization:** Biruk Beletew Abate.

**Data curation:** Biruk Beletew Abate.

**Formal analysis:** Biruk Beletew Abate.

**Funding acquisition:** Biruk Beletew Abate.

**Investigation:** Biruk Beletew Abate.

**Methodology:** Biruk Beletew Abate.

**Project administration:** Biruk Beletew Abate.

**Resources:** Biruk Beletew Abate.

**Software:** Biruk Beletew Abate.

**Supervision:** Biruk Beletew Abate, Teshome Gebremeskel Aragie.

**Validation:** Biruk Beletew Abate, Teshome Gebremeskel Aragie.

**Visualization:** Biruk Beletew Abate, Teshome Gebremeskel Aragie, Getachew Tesfaw.

**Writing – original draft:** Biruk Beletew Abate, Teshome Gebremeskel Aragie, Getachew Tesfaw.

**Writing – review & editing:** Biruk Beletew Abate, Teshome Gebremeskel Aragie, Getachew Tesfaw.

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
