## [Decision Letter · Decision Letter 0]

29 Jul 2020

PONE-D-20-09693

Magnitude of underweight, wasting and stunting among HIV positive children in East Africa: a systematic review and meta-analysis

PLOS ONE

Dear Dr. Biruk Beletew ,

Thank you for submitting your manuscript to PLOS ONE. After careful consideration, we feel that it has merit but does not fully meet PLOS ONE’s publication criteria as it currently stands. Therefore, we invite you to submit a revised version of the manuscript that addresses the points raised during the review process.

We look forward to receiving your revised manuscript.

Kind regards,

Claudia Marotta

Academic Editor

PLOS ONE

Journal Requirements:

2. We noticed you have some minor occurrence of overlapping text with the following previous publications, which needs to be addressed:

Rose AM, Hall CS, Martinez-Alier NAetiology and management of malnutrition in HIV-positive children Archives of Disease in Childhood 2014;99:546-551.

Eyasu Desta Menamo, 2014, Impact of Household Food Insecurity on Adherence to Antiretroviral Therapy (ART) among Urban PLHIV, Hamburg, Bedey Media GmbH, https://www.anchor-publishing.com/document/276651

Gabriel Anabwani, Peter Navario, Nutrition and HIV/AIDS in sub-Saharan Africa: An overview, Nutrition, Volume 21, Issue 1, 2005, Pages 96-99, ISSN 0899-9007, https://doi.org/10.1016/j.nut.2004.09.013. (http://www.sciencedirect.com/science/article/pii/S0899900704002254)

Byron, E., Gillespie, S., & Nangami, M. (2008). Integrating Nutrition Security with Treatment of People Living with HIV: Lessons from Kenya. Food and Nutrition Bulletin, 29(2), 87–97. https://doi.org/10.1177/156482650802900202

Trehan I, O'Hare BA, Phiri A, Heikens GT. Challenges in the Management of HIV-Infected Malnourished Children in Sub-Saharan Africa. AIDS Res Treat. 2012;2012:790786. doi:10.1155/2012/790786

Beletew, B., Mengesha, A., Wudu, M. et al. Prevalence of neonatal hypothermia and its associated factors in East Africa: a systematic review and meta-analysis. BMC Pediatr 20, 148 (2020). https://doi.org/10.1186/s12887-020-02024-w

The text that needs to be addressed involves the Background section.

In your revision ensure you cite all your sources (including your own works), and quote or rephrase any duplicated text outside the methods section. Further consideration is dependent on these concerns being addressed.

3. Thank you for inlcuding your funding statement; "no"

4. Thank you for including your competing interests statement; "no"

Additional Editor Comments (if provided):

Dear Authors,

I appreciate your manuscript, but need major revisions.

Following reviewer suggestions you can improve your paper.

Reviewers' comments:

Reviewer's Responses to Questions

**Comments to the Author**

1. Is the manuscript technically sound, and do the data support the conclusions?

Reviewer #1: Yes

Reviewer #2: Partly

2. Has the statistical analysis been performed appropriately and rigorously? 

Reviewer #1: Yes

Reviewer #2: Yes

3. Have the authors made all data underlying the findings in their manuscript fully available?

Reviewer #1: Yes

Reviewer #2: No

4. Is the manuscript presented in an intelligible fashion and written in standard English?

Reviewer #1: Yes

Reviewer #2: No

5. Review Comments to the Author

Reviewer #1: Authors wrote a very interesting manuscript on an important issue. Congratulations.

Only some suggestions:

1.Introduction: well wrote

2. Methods: "The search date was December 30/2019" is in yellow please remove the color

3. Results, Figure and Tables: I appreciate a lot

4. Discussion and Conclusion: if you can improve your discussion with some items:

- the definition on Children at risk. underweight, wasting and stunting among HIV positive children are the most vulnerable group and need more medical and research attention

(Marotta C, Di Gennaro F, Pizzol D, et al. The At Risk Child Clinic (ARCC): 3 Years of Health Activities in Support of the Most Vulnerable Children in Beira, Mozambique. Int J Environ Res Public Health. 2018;15(7):1350").

- The central role of task shifting to sharing experience and best practice

(Marotta C, Giaquinto C, Di Gennaro F, et al. Pathways of care for HIV infected children in Beira, Mozambique: pre-post intervention study to assess impact of task shifting. BMC Public Health. 2018;18(1):703. Published 2018 Jun 7.)

Reviewer #2: The first time you use an abbreviation, please spell its and then use the abbreviation (i.e. SAM, HIV)

Abstract

method: who HIV condition is not mentioned?

Results: why in the abstract do you go in deep regarding the period (2008-2014/ 2015-2019) and no data on malnutrition on HIV negative patients are reported?

Background

About the definition of under-nutrition (as part of malnutrition) authors should refer to WHO definition that is not that reported

If the authors focused the review in East Africa, the background should be focused also in East and not in sub-Saharan Africa (especially considering that they are suggesting a further review on Sub-saharan)

the whole background is quite confused with a mix of epidemiological data and possible explanation of interaction between HIV and malnutrition: please reorganise.

Methods

The Review question “What is the pooled prevalence of under-weight, wasting and stunting in East Africa context?” has no reference on HIV

The search strategy again does not contain the HIV term

Results

- “A total of 3094 studies were identified; 2050 from PubMed, 12 from Cochrane Library, 1010 from Google Scholar and 22 from other 120 sources. After duplication removed, a total of 970 articles remained” Are authors saying that PubMed database had more than 1000 duplicates?

Discussion

authors should discuss the mutual relationship between HIV and nutrition

why authors divided studies before and after 2014? They should discuss their choice

Authors should provide data on nutrition status among HIV negative children in the same context

Author should provide limitations and strength of this study (those provided are not enough especially limitations)

which clinical, political and social implications have these results?

English editing required

6. PLOS authors have the option to publish the peer review history of their article (what does this mean?). If published, this will include your full peer review and any attached files.

Reviewer #1: **Yes: **Francesco Di Gennaro

Reviewer #2: No

---

## [Author Response · Author response to Decision Letter 0]

8 Aug 2020

Date: Jul 29 2020 08:52AM

To: "PLOS ONE" plosone@plos.org

From: "Biruk Beletew" birukkelemb@plos.com

Subject: Submitting Revision manuscript [PONE-D-20-09693]

Magnitude of underweight, wasting and stunting among HIV positive children in East Africa: a systematic review and meta-analysis

Claudia Marotta

Academic Editor

PLOS ONE

Dear Editor and Reviewers we appreciate the careful feedback on our manuscript. Since we have agreed with all points you raised we believe we have carefully amended the paper as per your point of view. We described these changes in detail by point by point response below.

Editor Comment: 1. Please ensure that your manuscript meets PLOS ONE's style requirements, including those for file naming. The PLOS ONE style templates can be found at

Author response: we have amended the manuscript as per the PLOS ONE's style requirements using the PLOS ONE style templates as a guide.

Editor Comment: 2. We noticed you have some minor occurrence of overlapping text with the following previous publications, which needs to be addressed:

Rose AM, Hall CS, Martinez-Alier NAetiology and management of malnutrition in HIV-positive children Archives of Disease in Childhood 2014;99:546-551.

Eyasu Desta Menamo, 2014, Impact of Household Food Insecurity on Adherence to Antiretroviral Therapy (ART) among Urban PLHIV, Hamburg, Bedey Media GmbH, https://www.anchor-publishing.com/document/276651

Gabriel Anabwani, Peter Navario, Nutrition and HIV/AIDS in sub-Saharan Africa: An overview, Nutrition, Volume 21, Issue 1, 2005, Pages 96-99, ISSN 0899-9007, https://doi.org/10.1016/j.nut.2004.09.013. (http://www.sciencedirect.com/science/article/pii/S0899900704002254)

Byron, E., Gillespie, S., & Nangami, M. (2008). Integrating Nutrition Security with Treatment of People Living with HIV: Lessons from Kenya. Food and Nutrition Bulletin, 29(2), 87–97. https://doi.org/10.1177/156482650802900202

Trehan I, O'Hare BA, Phiri A, Heikens GT. Challenges in the Management of HIV-Infected Malnourished Children in Sub-Saharan Africa. AIDS Res Treat. 2012;2012:790786. doi:10.1155/2012/790786

Beletew, B., Mengesha, A., Wudu, M. et al. Prevalence of neonatal hypothermia and its associated factors in East Africa: a systematic review and meta-analysis. BMC Pediatr 20, 148 (2020). https://doi.org/10.1186/s12887-020-02024-w

The text that needs to be addressed involves the Background section.

In your revision ensure you cite all your sources (including your own works), and quote or rephrase any duplicated text outside the methods section. Further consideration is dependent on these concerns being addressed.

Author response: we have paraphrased the sentences which have textual overlaps. In addition we put sentences in quotation when we take others idea. Moreover, we cited those references we taken idea from. 

Editor Comment: 3. Thank you for including your funding statement; "no". Please clarify the sources of funding (financial or material support) for your study. List the grants or organizations that supported your study, including funding received from your institution.

Author response: The authors received no specific funding for this work

Editor Comment: 4. Thank you for including your competing interests statement; "no". Please complete your Competing Interests on the online submission form to state any Competing Interests. If you have no competing interests, please state "The authors have declared that no competing interests exist.", as detailed online in our guide for authors at http://journals.plos.org/plosone/s/submit-now

Author response: The authors have declared that no competing interests exist

Editor Comment: 5. In your Data Availability statement, you have not specified where the minimal data set underlying the results described in your manuscript can be found. 

 Author response: All relevant data are within the paper and its Supporting information files

Editor Comment: Dear Authors, I appreciate your manuscript, but need major revisions.

Following reviewer suggestions you can improve your paper.

Author response: thank you again for your constructive comments and for your optimism

Response to Reviewers

To Reviewer 1 (Francesco Di Gennaro)

Reviewer comment 1: Authors wrote a very interesting manuscript on an important issue. Congratulations.

Only some suggestions:

Authors’ response: thank you again for your constructive comments and for your optimism

Reviewer comment 1.Introduction: well wrote

Reviewer comment 2. Methods: "The search date was December 30/2019" is in yellow please remove the color

Authors’ response: remove as recommended

Reviewer comment 3. Results, Figure and Tables: I appreciate a lot

Authors’ response: thank you very much 

Reviewer comment 4. Discussion and Conclusion: if you can improve your discussion with some items:

- the definition on Children at risk. underweight, wasting and stunting among HIV positive children are the most vulnerable group and need more medical and research attention

(Marotta C, Di Gennaro F, Pizzol D, et al. The At Risk Child Clinic (ARCC): 3 Years of Health Activities in Support of the Most Vulnerable Children in Beira, Mozambique. Int J Environ Res Public Health. 2018;15(7):1350").

- The central role of task shifting to sharing experience and best practice

(Marotta C, Giaquinto C, Di Gennaro F, et al. Pathways of care for HIV infected children in Beira, Mozambique: pre-post intervention study to assess impact of task shifting. BMC Public Health. 2018;18(1):703. Published 2018 Jun 7.)

Authors’ response: we really thank you for supplying us such sources for discussion and conclusion. We have amended these sections as per your recommendation and cited the references.

To Reviewer 2 

Reviewer comment: The first time you use an abbreviation, please spell its and then use the abbreviation (i.e. SAM, HIV)

Authors’ response: we used extended form of all abbreviations at their first use as per your recommendation. 

Abstract

Reviewer comment: method: who HIV condition is not mentioned?

Authors’ response: among HIV positive children in East Africa

Reviewer comment: Results: why in the abstract do you go in deep regarding the period (2008-2014/ 2015-2019) and no data on malnutrition on HIV negative patients are reported?

Authors’ response: We have revised the method in the abstract removing unnecessary details as per your recommendation. Regarding data on malnutrition on HIV negative patients, all the included studies have done on HIV positive patients. As a result we haven’t extracted data on malnutrition among HIV negative patients since the study population of included studies are HIV positive children. 

Background

Reviewer comment: About the definition of under-nutrition (as part of malnutrition) authors should refer to WHO definition that is not that reported

Authors’ response: we have incorporated WHO definition of under-nutrition (as part of malnutrition) as “According to WHO malnutrition, in all its forms, includes under-nutrition (wasting, stunting, underweight), inadequate vitamins or minerals, overweight, obesity, and resulting diet-related non communicable diseases(1, 2).”

Reviewer comment: If the authors focused the review in East Africa, the background should be focused also in East and not in sub-Saharan Africa (especially considering that they are suggesting a further review on Sub-saharan)

the whole background is quite confused with a mix of epidemiological data and possible explanation of interaction between HIV and malnutrition: please reorganise.

Authors’ response: we paraphrased the background as per your comment. We were trying to show the burden from global to local (global – Africa-Sub-Saharan-East Africa) since we have been commented to write it as such. To address your comment we have added data in Eas Africa context.

Reviewer comment: Methods

The Review question “What is the pooled prevalence of under-weight, wasting and stunting in East Africa context?” has no reference on HIV

Authors’ response: we have amended the research question considering your comment as “What is the pooled prevalence of under-weight, wasting and stunting among HIV positive children in East Africa context?”

Reviewer comment: The search strategy again does not contain the HIV term

Authors’ response: Sorry it was typo error, we missed HIV and AIDS during write up of the manuscript. However we used those terms during searching. Now we have revised the search strategy including the term HIV and AIDS as we actually searched.

Results

Reviewer comment: - “A total of 3094 studies were identified; 2050 from PubMed, 12 from Cochrane Library, 1010 from Google Scholar and 22 from other 120 sources. After duplication removed, a total of 970 articles remained” Are authors saying that PubMed database had more than 1000 duplicates?

Authors’ response: the duplicates are not only PubMed database rather the duplicates are from all databases. After collecting studies from all database in endnote library we DEduplicated (removed duplicated studies) in the endnote itself using the following steps. First Edit- preferences-- duplicate –tick all boxes containing author ,year, title……-Apply then –Ok.

Then we gone to reference-Find Duplicate –then we compared the duplicate side by side. Finally we removed duplicates.

Discussion

Reviewer comment: authors should discuss the mutual relationship between HIV and nutrition

Authors’ response: we have revised the discussion as per your and the other reviewers comment. The relationship between HIV and nutrition discussed at the last paragraph of the discussion.

Reviewer comment: why authors divided studies before and after 2014? They should discuss their choice

Authors’ response: we want to assess the trend and the number of studies before and after 2014 is almost balanced to compare the change in magnitude with period changes.

Reviewer comment: Authors should provide data on nutrition status among HIV negative children in the same context

Authors’ response: study subjects of all included studies were HIV positive children. As a result we assessed the pooled nutritional status of HIV positive children. As you said it would be helpful if comparative studies are performed and meta-analysis of those comparative studies is recommended by authors of this study. We have included this in the limitation of the study section of the manuscript.

Reviewer comment: Author should provide limitations and strength of this study (those provided are not enough especially limitations)

Authors’ response: We have now expanded the limitations of the study as per your recommendation as “there are a few limitations to consider in the present study. First, due to the cross-sectional design, the observed results cannot be interpreted as causal. Second, the self-reported measures of variables are subject to measurement, self-report, social desirability, and recall biases. Third, publication bias may occur because all grey literature may not be included and language bias; since all included studies are published in English only. Forth, since only HIV positive children were taken as a study subjects it is difficult to present the result comparing with HIV negative children.”

Reviewer comment: which clinical, political and social implications have these results?

Authors’ response: we have included clinical, political and social implications of the study. “Health professionals working with HIV positive children should routinely screen and manage under nutrition (underweight, wasting and stunting) (clinical implication). Policy makers should incorporate strategies regarding prevention, screening and management of under nutrition in management of HIV/AIDS (political implication). Parents of HIV positive children should improve their feeding practice so as to prevent under nutrition (social implication).”

Reviewer comment: English editing required

Authors’ response: We have consulted native English-speaking colleagues and they have helped editing the paper. We (all authors) have also edited it through repetitive checking and online grammar editor.

---

## [Decision Letter · Decision Letter 1]

17 Aug 2020

Magnitude of underweight, wasting and stunting among HIV positive children in East Africa: a systematic review and meta-analysis

PONE-D-20-09693R1

Dear Dr. Biruk Beletew,

We’re pleased to inform you that your manuscript has been judged scientifically suitable for publication and will be formally accepted for publication once it meets all outstanding technical requirements.

Kind regards,

Claudia Marotta

Academic Editor

PLOS ONE

Additional Editor Comments (optional):

Dear Authors,

you have written a very interesting article

Congratulations

Reviewers' comments:

Reviewer's Responses to Questions

**Comments to the Author**

1. If the authors have adequately addressed your comments raised in a previous round of review and you feel that this manuscript is now acceptable for publication, you may indicate that here to bypass the “Comments to the Author” section, enter your conflict of interest statement in the “Confidential to Editor” section, and submit your "Accept" recommendation.

Reviewer #1: All comments have been addressed

Reviewer #2: All comments have been addressed

2. Is the manuscript technically sound, and do the data support the conclusions?

Reviewer #1: Yes

Reviewer #2: Yes

3. Has the statistical analysis been performed appropriately and rigorously? 

Reviewer #1: Yes

Reviewer #2: Yes

4. Have the authors made all data underlying the findings in their manuscript fully available?

Reviewer #1: Yes

Reviewer #2: Yes

5. Is the manuscript presented in an intelligible fashion and written in standard English?

Reviewer #1: Yes

Reviewer #2: Yes

6. Review Comments to the Author

Reviewer #1: Authors improved their manuscript. the interaction between authors and reviewers was effective and useful in order to improve the manuscript and give the scientific community a very relevant article on a highly topical issue. I think that can be accepted

Reviewer #2: (No Response)

7. PLOS authors have the option to publish the peer review history of their article (what does this mean?). If published, this will include your full peer review and any attached files.

Reviewer #1: **Yes: **Francesco Di Gennaro

Reviewer #2: No

---

## [Editor Report · Acceptance letter]

8 Sep 2020

PONE-D-20-09693R1 

Magnitude of underweight, wasting and stunting among HIV positive children in East Africa: a systematic review and meta-analysis 

Dear Dr. Abate:

I'm pleased to inform you that your manuscript has been deemed suitable for publication in PLOS ONE. Congratulations! Your manuscript is now with our production department. 

Kind regards, 

on behalf of

Dr. Claudia Marotta 

Academic Editor

PLOS ONE